# iPSC Preparation and Epigenetic Memory: Does the Tissue Origin Matter?

**DOI:** 10.3390/cells10061470

**Published:** 2021-06-11

**Authors:** Giuseppe Scesa, Raffaella Adami, Daniele Bottai

**Affiliations:** Department of Health Science, University of Milan, via A di Rudinì 8, 20142 Milan, Italy; giuseppe.scesa@unimi.it (G.S.); raffaella.adami@unimi.it (R.A.)

**Keywords:** iPSCs, reprogramming methods, Yamanaka factors, epigenetic memory, methylation

## Abstract

The production of induced pluripotent stem cells (iPSCs) represent a breakthrough in regenerative medicine, providing new opportunities for understanding basic molecular mechanisms of human development and molecular aspects of degenerative diseases. In contrast to human embryonic stem cells (ESCs), iPSCs do not raise any ethical concerns regarding the onset of human personhood. Still, they present some technical issues related to immune rejection after transplantation and potential tumorigenicity, indicating that more steps forward must be completed to use iPSCs as a viable tool for in vivo tissue regeneration. On the other hand, cell source origin may be pivotal to iPSC generation since residual epigenetic memory could influence the iPSC phenotype and transplantation outcome. In this paper, we first review the impact of reprogramming methods and the choice of the tissue of origin on the epigenetic memory of the iPSCs or their differentiated cells. Next, we describe the importance of induction methods to determine the reprogramming efficiency and avoid integration in the host genome that could alter gene expression. Finally, we compare the significance of the tissue of origin and the inter-individual genetic variation modification that has been lightly evaluated so far, but which significantly impacts reprogramming.

## 1. iPSCs Production

The derivation of induced pluripotent stem cells (iPSCs) from somatic human cells by Takahashi and Yamanaka in 2007 [1] represented a turning point for the field. For the first time, they provided isogenic pluripotent cells with the potential for personalized cell replacement therapies; no ethical issues would be created by using the somatic cells. This opportunity marks a decisive step compared to the generation of human ESCs arranged by Thomson et al. in 1998 [2], where they described the derivation of pluripotent stem cells from the human blastocyst and depicted their characteristics such as retention of normal karyotypes, high expression levels of telomerase activity, high proliferation, and differentiation capability.

The 2012 Nobel prize sanctioned that specialization of cells is reversible and that adult somatic cells could be reprogrammed to an immature, pluripotent state. However, several years after this breakthrough, the comparison of the characteristics of iPSCs with ESCs made clear that *not everything that glitters is gold*. Since then, many efforts have been made to better understand the biological peculiarities of iPSCs and make the best use of those cells.

Takahashi and Yamanaka [1,3] were able to generate iPSCs using retroviral transduction into adult somatic cells. The process involved four transcription factors: (a) octamer-binding transcription factor 4 (Oct4), (b) sex-determining region Y-box 2 (Sox2), (c) Kruppel Like Factor 4 (KLF4), and (d) the oncogene c-MYC; all four are called OSKM. These were selected after testing many genes supposedly involved in the first stages of ESCs’ development [3].

Since then, numerous different methods have been established to improve reprogramming efficiency. They included varying the source of cells to be reprogrammed (Figure 1A), the genes used for the reprogramming, and the methods of reprogramming (Figure 1B). Takahashi’s retroviral approach reached an efficiency of 0.02% in reprogramming [3], while further attempts by other groups achieved an efficiency of 0.05–0.08% [4].

These changes in reprogramming methods merged approaches still based on integration into the host genome of exogenous reprogramming factors, and included lentiviruses and transposons as vectors. Those methods offer higher reprogramming efficiency, although the integration process entails permanent DNA modification, potential insertional mutagenesis, and transgenes reactivation.

To overcome these and other potential pitfalls that could limit the future use of iPSCs for tissue regeneration, additional methods were developed based on viral and non-viral, non-integrating delivery of the transgenes containing OSKM [5,6,7,8,9,10]. Further modification of the design of the reprogramming expression vectors and new methods of delivery were designed to minimize or eliminate vector sequences that could be integrated into the iPSCs genome [11,12]. However, these different approaches resulted in a significant decrease, in some cases more than 100-fold, of the reprogramming efficiency [11].

The following sections present the induction methods in two groups: viral and non-viral (Figure 1B).

### 1.1. Viral Reprogramming Methods

Within the viral approaches, apart from the aforementioned genome-integrating vectors, we can include Adeno (AV), Adeno-associated (AAV), and Sendai viruses (SV).

Viral non-integrating reprogramming methods were developed to overcome concerns related to exogenous gene integration and DNA modification at the expense of generally lower reprogramming efficiency. For example, the Adenoviral approach permits an efficiency of only 0.001–0.0001% in mouse fibroblasts [13] and 0.0002% in human fibroblasts [14]; multiple infections might be required [15]. In addition, the use of viral vectors might elicit an immune response in the host after cell transplantation, thus compromising the efficacy of the therapy [16].

#### 1.1.1. Sendai Virus

Sendai virus is a negative sense, mRNA virus belonging to Paramixoviridae family [17]. It is non-pathogenic to humans, and its use as a viral vector has several advantages: (1) being an mRNA virus, it does not enter the nucleus in its lifecycle, thus eliminating the risk of modifying the host genome and/or causing gene silencing by epigenetic changes [18]; (2) it shows a broad tropism, being able to infect several cell types in vitro [19,20,21,22,23]; (3) due to its non-integrating nature, viral genome is diluted to every cell duplication, allowing its removal from the reprogrammed cells; and (4) it allows the production of a large number of proteins, thus allowing multiplicity of infection (MOI) reduction. Sendai viral vectors were successfully used to reprogram fibroblast cells [24], as well as blood [25,26] and renal epithelial cells in the urine [27]. This technique is quite efficient, ranging between 0.01% and 4% in the generation of human iPSCs at 25 days of induction [20,24,25,28,29]. Up to 10 passages or a high-temperature culture (39 °C) might be necessary to remove the viral genome completely [11]; however, an auto-erasable, replication-deficient Sendai virus was recently developed using microRNA-302 which impedes viral replication by blocking the viral RNA-dependent RNA polymerase [30].

#### 1.1.2. Adenovirus

Adenovirus is a non-integrating virus [31] that remains in the epichromosomal form in all cell types, except in egg cells [32]. Adenovirus offers a large cargo capacity, a transient expression, and rapid clearance from dividing cells, thus requiring multiple rounds of infection. The reprogramming efficiency is low, 0.001–0.0001% in mouse cells and 0.0002% in human cells, most likely because of the low infection efficiency and the narrow expression window of reprogramming factors [16].

#### 1.1.3. Adeno-Associated Virus

Adeno-associated virus is a non-pathogenic, non-autonomous single-stranded DNA virus, unable to replicate without the presence of a co-infecting helper virus. In its absence, AAV’s genome remains in episomal form within the infected cells, although integration into the host’s genome was reported in fewer than 10% of cases. AAVs are associated with weak immune reactions, primarily deriving from innate responses to the viral genome [33,34]. For all these reasons, this vector has been used in more than 100 clinical trials [35]. However, the need for multiple rounds of transduction for cell reprogramming, limited transgene capacity (5 kb), and low efficiency (less than 0.01%) [16,35] still limit AAVs’ use as a vector for inducing pluripotency.

### 1.2. Non-Viral Reprogramming Methods

Along with these techniques, we can include mRNA transfection, miRNA infection/transfection, PiggyBac, Episomal plasmids/vectors, minicircle vectors, and protein and chemical induction [11,31].

#### 1.2.1. mRNA Transfection

mRNA transfection was first used for cell reprogramming by Warren’s group. They overcame several obstacles to transcribe mRNAs to express reprogramming factors efficiently, reaching a 1.4% reprogramming efficiency [36]. Moreover, the addition of Lin28 to the Yamanaka reprogramming factor protocol, valproic acid in the cell culture medium, and a change of O_2_ concentration to 5% allowed for an increase in efficiency to 4.4% [36]. The need for daily additions of mRNA to the cells limited to fibroblast make this method less appealing than other approaches [11].

#### 1.2.2. miRNA Infection/Transfection

Regarding the miRNA infection/transfection, several miRNA, such as miR-302b or miR-372, are strongly expressed in ESCs. Their addition to Yamanaka factors increased up to 15-fold reprogramming efficiency for OSKM alone [37]. Interestingly, some miRNAs could reprogram cells at high efficiency even in the absence of co-transfection with OSKM, bringing the efficiency of reprogramming for BJ-1 fibroblasts up to 10% [38].

#### 1.2.3. Transposons: PiggyBac and Sleeping Beauty

These two transposons usually consist of a polycistronic transcript containing the OSKM factors joined by 2A peptides, allowing post-translational cleavage of the polyprotein into single reprogramming proteins as well as maintenance of their stoichiometric co-expression [39]. PiggyBac is a transposon, a mobile genetic element, that can be easily inserted and removed from chromosomal TTAA sites in the genome. Using a transposase, it can be integrated and subsequently excised from the genome [11]. When the OSKM factors are cloned into the PiggyBac vector and co-transfected into mouse embryonic fibroblasts (MEFs), reprogramming efficiency ranged from 0.02 to 0.05% of the total transfected cells [40,41]. This technique requires only a single transfection; the transposon can transport substantial cargo and presents low immunogenicity.

An intrinsic feature of the PiggyBac vector is its integration into the host genome. However, it could be cleanly excised from the iPSCs genome. Potential reintegration is conceivable due to the use of the same enzyme for insertion and excision. This reintegration risk forces a tight screening of iPSC clones to confirm the absence of integration and is time-consuming. As previously mentioned, reprogramming efficiency is quite low (0.01–0.1%) but can be improved using sodium butyrate. Regardless, the efficiency remains 50-fold lower than retroviral-mediated reprogramming methods [42].

Sleeping Beauty transposon vector differs from PiggyBac in its ability to integrate randomly into host genomes, thus showing no integration tendency with regard to specific genes and gene-regulatory elements [43].

#### 1.2.4. Episomal Plasmids

Episomal plasmids ensure a technically simple procedure, a stable transgene expression due to their self-replication, and a low immunogenicity, allowing their removal by culturing cells in the absence of drug selection [44]. Due to their vulnerability to exonucleases, episomal vectors have an extremely low reprogramming efficiency, primarily due to the short expression time in the cells. It is possible to overcome this issue by repeating transfections daily; however, the reprogramming efficiency remains unsatisfactory (0.0003–0.0006%) [45]. The inclusion of NANOG, LIN28, and LT (SV40 large T gene) as reprogramming factors enhanced the efficiency 100 times, making it comparable to viral-based methods [46]. Instead of using a single plasmid for every Yamanaka factor, which is laborious and less efficient, as only a few cells receive all the plasmids, some groups use polycystronic plasmids to obtain a “3+1 delivery” of the reprogramming factors (with Oct4, Klf4, and Sox2, carried by one plasmid, and c-Myc by the other) [15], while other groups rely on one single plasmid to deliver all four genes [47], under the control of a constitutively active CAG promoter. However, these last methods do not ensure an adequate stoichiometric co-expression of the multiple reprogramming factors. The use of picornaviral 2A self-cleaving peptides to link reprogramming factors, when used as a polycistronic construct [48], can partially ameliorate the balance of the expression of the four genes. However, there is still the chance that polycistronic plasmid could produce an unbalanced expression of each reprogramming factor [49]. This issue, together with the large size of the plasmid, hampers the efficiency of plasmid-based reprogramming [16].

#### 1.2.5. Minicircle Vectors

Minicircle vectors are supercoiled DNA episomal vectors similar to a standard plasmid but containing only the eukaryotic promoter and the cDNA(s) of the genes to be expressed. Their small size, resistance to cleavage, extremely low immunogenicity, and high transfection efficiency make them a good tool for cell reprogramming [16], despite a very low reprogramming efficiency (0.005%) and long reprogramming time (14–18 days). Thus, multiple rounds of transfection are required, causing a reduction in cell viability [50]. In order to improve the efficiency of reprogramming, various researchers used electropulsation, included additional reprogramming factors and/or microRNAs, used small molecules, and included hypoxic conditions [51,52,53,54]. Tight screening of the clones is necessary to exclude the integration of transgene sequences.

#### 1.2.6. Liposomal Magnetofection

Liposomal magnetofection is a non-viral technique that allows the delivery of nucleic acids in cultured cells by mixing nucleic acid and magnetic nanoparticles in cationic lipids. The lipids are concentrated on the surface of the cells using a magnetic field [55]. This technique has little chance of genomic integration, requires only a single transfection, and has low immunogenicity. There are rare cases of genomic integration. Consequently, screening iPSC clones is necessary to confirm the absence of integration [56]. Moreover, its reprogramming efficiency is between 0.032% and 0.040% after 8 days [16].

#### 1.2.7. Proteins

Although the bioactive forms of reprogramming proteins can be synthesized by prokaryotic or eukaryotic systems, the main hurdle for reprogramming is their limited capability to cross the cell membrane. To overcome this obstacle, the protein approach takes advantage of the HIV-TAT protein (a protein transduction domain) in delivering recombinant proteins. This technique allows the introduction of proteins into cells from the external environment without permeabilization agents [57]. The efficiency of this procedure is quite low, at 0.006% of mouse fibroblasts [58] and 0.001% of human fibroblasts [9,59]. To improve the reprogramming efficiency, some authors supplemented the culture media with valproic acid (VPA), with 0.006% of cells induced to pluripotency after 30–35 days [58].

Protein transduction domains later became a method to deliver not only proteins but also other macromolecules. Those included peptide nucleic acids (PNA), antisense, short-interfering ribonucleic acids (siRNA), liposomes, iron nanoparticles, and plasmids [57].

#### 1.2.8. Chemical Induction

Despite the methods mentioned above for reprogramming, the highest degree of safety at the cost of low reprogramming efficiency is represented by iPSC generation through the use of small molecules to obtain chemically induced pluripotent stem cells (CiPSCs).

Hou et al. developed a combination of six small molecules (obtained after an intense work of screening of more than 10,000 compounds). They included several cAMP agonists (Forskolin, Rolipram, and Prostaglandin E2) and epigenetic modulators (sodium butyrate, 3-deazaneplanocin A (DZNep), 5-Azacytidine, and RG108) to generate chemically induced iPSCs (CiPSCs). Interestingly, they found that small molecule (sm) iPSCs could be generated using only one gene of the OSKM, namely Oct4, with the addition of CHIR99021, tranylcypromine (VC6T), VPA, and 616452 [8,60]. Compared to ESCs, CiPSCs have similar doubling time, gene expression profiles and differentiation ability, and they generate teratomas and chimeric mice [8,61]. Moreover, it is intriguing that different chemical cocktails are needed to induce other source cells [60]. To date, current chemical reprogramming efficiency is only 0.2%, with an induction time of more than 36 days that was recently reduced to 16 days [62,63].

## 2. Comparison between Different Methods of Reprogramming

The presence of the high level of copy number variation (CNV) in hiPSCs compared to hESCs or human somatic cell samples can be explained with two, not self-excluding, hypotheses: (a) they are gained de novo during the reprogramming procedure or in vitro iPSCs culture or, (b) they are present in the starting somatic cell population that could also be a mosaic [64,65,66]. Since the first work of Yamanaka [3], many efforts have been made to understand how reprogramming could impact the quality of iPSCs.

In a study conducted by Ma et al., the comparison between different methods of reprogramming (i.e., Sendai virus (IPSCs-S) and retroviral (iPSCs-R) methods) indicated that some lines, such as iPS-S4, iPS-S5, and iPS-R2, did not display significant genomic macroscopic alterations. Copy number variation (CNV) analysis did not entirely exclude the presence of small insertion-deletions (indels), point mutations, or translocations [67].

In other papers, the genetic stability of independent iPSCs lines with common donors was tested by CNV SNP microarrays [65]. However, lines produced using integrating vectors showed a trended but not significantly higher frequency of clinically significant CNV (58%) compared with non-integrating vectors (41%). Since this study compared iPSCs lines obtained from the same donor, the authors could evaluate whether the CNV differences were due to the tissue of origin or the method or reprogramming [65]. Similarly, Schlaeger et al., [29] compared episomal vector reprogramming, Sendai virus, RNA, and lentivirus reprogramming, finding no differences in CNV. Many different groups found that if differences do exist between the reprogramming methods, these are most likely present when the reprogramming is made using integrating viral vectors; they are also very subtle, although they could be more deleterious [68,69,70,71].

Taken together, these data suggest that different induction methods do not contribute significantly to the genic alterations found in iPSC lines obtained from isogenic cells; most likely, the genic impairment found in iPSCs could be ascribed to the somatic donor cells or the cultivation time.

## 3. Source Cells for Reprogramming

The cells used for reprogramming depend on the organism, the availability of the tissue, and the kind of differentiated cells we would like to produce from the iPSCs. As addressed in the following sections, many tissues cannot be used because they are unavailable (i.e., brain tissue) unless obtained as discarded tissue.

However, to date, iPSCs have been obtained using a plethora of tissues (Figure 1B). In this regard, some of these methods require invasive procedures such as biopsies, as in primary skin fibroblasts. More accessible sources are available, namely peripheral blood from which we can retrieve T cells [25], B cells [72], hematopoietic stem cells [73], and bone marrow cells. Recently, iPSCs have been produced by choosing less invasive cells to obtain, such as keratinocytes isolated from hair follicles [74]. Very often, cell sources have been obtained from biological waste material [75]. Examples include renal epithelial cells in the urine [27,76], mesenchymal stem cells from teeth and fat tissue [77], liver and stomach cells [78], melanocytes [79], neural stem and progenitor cells [80], and embryonic and extraembryonic tissue [81]. These outcomes indicate that cells of all tissues might be converted into iPSCs. The final point about the origin of source cells concerns their age. Senescent cells or cells obtained from the elderly are induced to iPSCs with more difficulties. However, Lapasset et al. found that their induction efficiency could be increased using a six-factor reprogramming cocktail (SOX2, OCT4, KLF4, NANOG, LIN28, and c-MYC) instead of the usual OSKM, which also eliminates the marks of cellular aging [82,83].

Depending on the cells of origin and the methods of reprogramming, gene cocktails other than OSKM, such as p53 shRNA, Lin28, L-Myc, SV40LT, Nanog, Glis1, and others, have been used in different reprogramming mixes, sometimes improving the efficiency of reprogramming in particular subsets of tissue sources [27,84].

An important aspect that we mentioned earlier is the presence of mosaicisms in the source cells that could negatively affect reprogrammed cells. The production of iPSCs from a patient affected by Down syndrome showed that the patient was a mosaic, since one-third of the reprogrammed cells were euploid, whereas the remaining 66% were trisomic [85]. Abyzov et al. demonstrated that 50% of the CNVs identified in the hiPSC lines were detectable, even at a very low frequency, in the source fibroblast population [66], indicating the presence of somatic mosaicism in these cells.

## 4. Epigenetic Alteration after Reprogramming

Independently of the reprogramming method, profound modifications of the epigenetic landscape of the donor cells appear during iPSC induction. Pluripotent stem cells [86] such as ESCs show a distinctive epigenetic profile, with active chromatin modifications. Histone acetylation, hypomethylated DNA, a tri-methylation at the 4th lysine residue of the histone H3 (H3K4me3), and tri-methylation at the 36th lysine residue of the histone H3 (H3K36me3) [87,88] locate primarily within the regions of genes responsible for pluripotency (Figure 2).

The opposite happens in tissue-specific genes [87]. Another interesting aspect of pluripotent stem cells is that they have elevated levels of the so-called bivalent domains, with methylation in H3K27me3 and H3K4me3 in differentiation-related genes, which can be easily activated or silenced, eliminating H3K27me3 or H3K4me3. This sensitive equilibrium is pivotal for the maintenance of stemness [87,89,90] (Figure 2).

During reprogramming, silencing of somatic cell genes and activation of pluripotency-associated genes are observed (Figure 1C), and they push for a de-differentiation of the cell into a naïve, pluripotent state. These cells are ultimately characterized by unlimited cell proliferation and differentiation into cells derived from the three germ layers in vitro; in vivo, they can form teratomas, generate chimeras, and complete organisms through tetraploid complementation. These are the most rigorous criteria for pluripotency characterization of pluripotent stem cells [91], which can be addressed only in non-human cells [92].

To date, the molecular mechanisms that underlie the derivation and maintenance of iPSCs are not yet wholly understood. Most of the studies on the transcriptional and epigenetic circumstances driving pluripotency and reprogramming have been performed on mice. Due to the strong cross-species similarities, many of these results have been translated to humans.

The change in gene expression occurs progressively due to a defined sequence of cellular and molecular events (Figure 2). It can be divided into initiation, maturation, and stabilization. Some of the phases concern cellular dynamics in which there is a change in cell size, a mesenchymal-to-epithelial transition, a change in proliferation rate, and a metabolic switch; the stabilization phase is transgene independent. Another aspect concerns transcription dynamics in which the somatic genes are switched off in the initiation phase. At the same time, cell cycle genes are activated from the maturation to the stabilization phase; the pluripotency genes are also activated from the maturation phase. Meanwhile, the epidermis genes are on only in the maturation phase. The epigenetic dynamic H3K4me2 (permissive) and H3K9me3 (repressive) methylation are turned on early in the activation phase and decrease along with the other phases. H3K4me3/H3K27me3 (bivalent) increases from the maturation phase throughout the stabilization phase, similar to DNA hydroxymethylation and histone acetylation. Finally, DNA methylation and demethylation increase during the stabilization phase [87,88].

## 5. Similarities and Differences between iPSCs and ESCs

Initially, iPSCs were considered very similar to ESCs, given their similarity in morphology, proliferation, marker expression, differentiation potential, and teratoma formation [3]. However, this resemblance was soon questioned, as differences started to be discovered in the pluripotent cells’ gene expression profiles. It was postulated that iPSCs might retain donor cell-specific transcriptome, along with DNA methylation states [93,94]. In their 2009 original work, Marchetto et al. compared the gene expression profiles of human neural stem cells (NSCs), NSC-derived iPSCs, and ESCs. Their transcriptome analysis revealed that, despite the similarities, iPSCs had insufficient repression of NSCs-associated genes, as well as insufficient induction of embryonic-specific genes.

Interestingly, the authors identified a group of upregulated genes in iPSCs that were silenced in both ESCs and NSCs, and postulated that they might be downstream genes involved in the reprogramming process [93]. Retention of such “epigenetic memory” appears to be due to limitations in reprogramming efficiency when compared to nuclear transfer-derived ESCs (NT-ESCs). iPSCs displayed an epigenetic profile that reflected their origin, while NT-ESCs turned out similar to ESCs [94]. Thus, current reprogramming methods cannot erase the methylation profile as efficiently as nuclear transfer [67].

In their pivotal work, Ma et al. compared four NT-ESCs derived from fetal human dermal fibroblasts (HDFs). Seven iPSCs cell lines were derived from the same HDFs using retroviral vectors (two lines), Sendai vectors (five lines), and two in vitro fertilized (IVF) ESC cell lines from the same egg donor were used for somatic cell nuclear transfer. The macroscopical analysis showed that the different pluripotent stem cells maintained similar morphology, expressed the same pluripotency markers, formed teratomas, and retained diploid karyotypes, with no detectable numerical or structural chromosomal abnormalities [67]. However, in the study mentioned above, some subchromosomal aberrations were observed in iPSCs and NT-ESCs. The iPSCs, NT-ESCs, and IVF ESCs bore an average of 1.8, 0.8, and 0.5 CNVs per line, respectively, that were not significantly different [67].

For the methylation, the NT-ESC profiles were more similar to IVF ESCs than iPSCs. The last retained eight-fold more epigenetic memory of the fetal origin HDF than NT-ESCs [67].

Many of the differentially methylated regions (DMRs) found in iPSCs derive from aberrant methylation events occurring during reprogramming, as shown in a paper by Lister et al. [95]. Like Ma et al. [67], the authors compared whole-genome profiles of iPSC generated with different methods (retroviral and episomal, non-integrating vectors), ESCs, somatic cells, and differentiated iPSCs. Their findings showed the presence of iPSCs-specific DMRs among other iPSCs (Figure 3), suggesting the existence of genomic regions more prone to aberrant methylation during the reprogramming process [95].

An exciting point that arose from Lister’s study is that the reprogramming of a somatic cell to a pluripotent state produces hundreds of aberrantly methylated loci located primarily at CG islands [95]. Moreover, even if iPSCs have insufficient reprogramming capacity due to progenitor somatic cell methylation, iDMR induced differently from both the starting and ending points of the reprogramming are present [95]. In addition, some loci have a strong propensity to be insufficiently or aberrantly reprogrammed. Finally, memory CG-DMRs and induced iDMR are contemporarily transferred and conserved due to the differentiation of the iPSCs [95] (Figure 3).

The genes’ location within the genome might play a role in silencing specific genes during reprogramming, with gene density having a significant impact on the efficiency of the methylation process. Isolated genes were found to be less methylated than more clustered genes. This might represent a better substrate for the cellular silencing machinery [96,97]. Among the DMRs present in many reprogrammed cell lines compared to H1-ESCs, centromeric and telomeric chromosomal portions presented hypomethylation, further supporting the notion that somatic cell reprogramming may be sensitive to DNA methylation alteration in these regions [95].

Histone H3 lysine 9 three-methylation (H3K9me3) (with repressive activity) was significantly enriched in iPSCs and located in the analogous area of the non-CG mega-DMRs, which are hypomethylated and not present in H1-ESCs [95]. These newly acquired methylated regions can be passed through generations to the differentiated cells, thus posing concerns for their clinical use and differentiation potential. Various iPSCs lines differentiated into trophoblast presented more than 210 differently methylated (140 hypomethylated and 70 hypermethylated) genes compared to ESCs. These hypermethylated and hypomethylated regions were transmitted through the trophoblast differentiation process with 88% and 46%, respectively, and are still present in iPSCs but not in the trophoblasts obtained from H1-ESCs [95].

The presence of DMRs has a potential effect on the phenotype of reprogrammed cells and a fundamental impact on pluripotency induction. C9orf64 is a conserved gene that does not present a protein domain and whose function is not yet depicted [98]. Together with the delta-like homolog 1 gene and the type III iodothyronine deiodinase gene (Dlk1-Dio3) [99] gene clusters, C9orf64 was shown to affect iPSC generation and pluripotency acquisition. In a paper by Ohi et al., the authors compared the transcriptome of ESCs, iPSCs, and their somatic progenitors. They found a tight correlation between higher expression levels and lower promoter methylation of C9orf64, TRIM4 (tripartite motif-containing 4), and catechol O-methyltransferase (COMT) [98]. When a meta-analysis of transcriptome generated by different groups was performed, C9orf64 was constantly found among the less silenced genes during iPSCs generation [98]. Interestingly, silencing of C9orf64 caused a decrease in mature iPSCs colonies, suggesting an active role of the somatic gene(s) for the reprogramming [98].

Similar outcomes were described in a study showing an analogous result at the proteome level. In contrast, the differences between ESCs and iPSCs could be depicted only when more biological replicated were added, most likely because this variability was introduced by the inclusion of different donors [100].

The expression of somatic genes is required to acquire full pluripotency during reprogramming, as shown by Stadtfeld et al. [99]. The comparison of mouse ESCs and iPSCs from identical genetic backgrounds indicated that the mRNA and micro mRNA were almost identical, except for a limited number of transcripts encoded within the cluster of imprinted genes, such as the Dlk1-Dio3. This cluster is located on distal mouse chromosome 12 that is crucial for proper fetal development. In fact, imprinting errors of this locus are mechanistically linked to developmental disorders such as Kagami-Ogata Syndrome and Temple Syndrome, but may also be implicated in respiratory diseases and cancer [101]. Imprinting of the Dlk1–Dio3 cluster is regulated by DMRs epigenetically modified during the germline phase of development. These include an intergenic DMR (IG-DMR) and intragenic differential methylation in the Glt2 promoter [102,103]. The fibroblasts differentiated from iPSCs that acquired a silenced paternal methylation state did not activate after the induction of the differentiation induced by valproic acid [99]. Very interestingly, ESCs (containing the OKSM transgene) and iPSCs derived from MEFs from the same genetic background were compared for their capacity to support the development of entirely iPSC-derived animals (all-iPSC mice) using 4n embryo complementation, a technique in which 2n pluripotent cells are injected into tetraploid (4n) blastocyst. The resulting 2n–4n chimeric embryos will develop normally, with the extraembryonic tissues composed of 4n cells, while the fetus will originate from the injected 2n ESCs [104,105,106]. ESC cell lines produced viable mice at the anticipated frequencies (13–20%), proving that the OKSM transgene does not invalidate embryo development. On the other hand, all the tested iPSC voided to do so [99].

## 6. Epigenetic Memory Biases iPSCs Differentiation towards the Cell of Origin

A necessary consequence of DMRs and, in general, of cellular epigenetic memory is their impact on iPSCs differentiation. Early iPSCs generation was primarily performed using fibroblasts as somatic, parental cells. Later successful reprogramming was conducted on cells obtained from different tissues, which allowed researchers to show a differentiation bias towards the lineage of the somatic cell of origin due to a significant difference in epigenetic patterns between parental cells [94,107]. Polo et al. compared iPSCs obtained from granulocytes (Gra) and skeletal muscle precursors (SMP) from the same mouse; some characteristic markers of Gra (Lysozyme and Gr-1) and Cxcr4 and Integrin B1 for SMP showed a higher level of expression in the respective cell of origin than in the iPSCs. This result was consistent in different experiments and, more importantly, within tissues obtained from different animals [107].

Assuming two-fold changes as the minimum difference between the expression, 1388 genes were differentially expressed between SMP-iPSCs and Gra-iPSCs, and 1090 genes were differentially expressed between splenic B cells (B)-iPSCs and tail tip–derived fibroblasts (TTF)-iPSCs. More importantly, the 100 genes with the highest level of expression within the different samples were grouped in the same cluster, indicating that iPSCs obtained from various isogenic somatic tissues were distinguishable in terms of gene expression [107].

Using restriction enzyme-based methylation analysis of promoters, Polo et al. were able to establish that very few loci were differentially modified by methylation. Moreover, methylation levels of the Lysozyme, Gr-1, Cxcr4, and Integrin B1 genes between SMP-iPSCs and Gra-iPSCs were not significantly different. This showed that DNA methylation differences are more understated than the changes in gene expression [107]. However, the promoters of the Cxcr4 and Itgb1 in SMPs and at the Lysozyme and Gr-1 in granulocytes showed a high level of acetylation in histone H3 and methylation (H3K4me3). This finding indicated that the expression differences previously described might be a consequence of the histone marks and, consequently, that iPSCs conserve an epigenetic memory of their cells of origin [107].

All those different transcripts provide functional differences in differentiation capabilities either in mouse cells [107,108] or human cells [109]. The effect on epigenetic memory on iPSCs differentiation tends to disappear through intensive culturing of the cells. Late-passage cells showed a more uniform differentiation potential than early passage cells, suggesting fluidity in the epigenetic landscape of iPSCs [107,108,109]. Hu et al. also found that, in human iPSCs, the differentiation to endothelial cells from exceedingly early (P < 10), early (10 < P < 20), and late (P > 20) passage iPSCs indicates that, at exceedingly early passages, the differentiation is more efficient in EC derived iPSCs than in fibroblast-derived iPSCs or cardiac progenitor cells iPSCs. However, late-passage cells exhibited minimal differences upon differentiation [109], in contrast with a previous study [110], most likely because, in the early work, the difference in passages was not high enough for epigenetic memory to be attenuated. Differences in expression profiles might affect differentiation potential and the survival and engraftment of differentiated iPSCs in vivo [109,111].

Regarding the capacity of differentiated iPSCs to engraft into damaged tissue, such as ischemic tissue, no difference was observed when endothelial cells differentiated from iPSCs derived from different tissues (endothelial cells, fibroblast, or cardiac progenitor cells). Nevertheless, when these implanted cells were retrieved from the tissues 14 days after transplantation, a larger percentage of endothelial markers was found in endothelial cells differentiated from endothelial-derived iPSCs [109].

Hargus et al. differentiated human iPSCs derived from fetal NSCs and dermal fibroblasts into neural precursor cells (NPCs) [112]. Whole-genome and brain-specific gene expression analysis revealed clear segregation of the different cell groups according to their somatic origin and retention of specific transcripts due to epigenetic memory. Interestingly, the authors assessed the functional effect of epigenetic memory of NPCs upon transplantation. However, the percentage of iPSCs-derived neurons was similar between the two groups after 5 weeks of in vitro differentiation. NPCs from fNSCs-iPSCs yielded a higher number and more differentiated neuronal cells than Fib-iPSCs-derived NPCs [112] when transplanted in cortex and striatum, indicating that the neural niche has more favorable effects for neuroectodermal than mesodermal iPSC-derived neurons.

On the other hand, many efforts have been made to search for the appropriate tissue of origin to produce iPSCs to be differentiated into mature cells of a particular lineage, either for physiological study or for transplantation. Neurons and cardiomyocytes have frequently been the subject of these studies.

In recent work, Wang et al. studied the epigenetic memory of five different retinal cell types that underwent reprogramming [113]. The authors determined whether r-iPSCs retain an epigenetic memory by selecting 33 lines (comprising iPSCs from 5 retinal cell types and fibroblast-derived iPSCs). A thorough study with different techniques including DNA methylation analysis, RNA-seq, and ChIP-seq of 8 histone marks was performed. Some progenitor genes including Nestin, Paired Box6, Sine Oculis Homeobox Homolog 3 and 6, Visual System Homeobox 2, and Meis Homeobox 1 showed DNA methylation profiles that were reset to the ESC state in almost all the iPSC lines. Some heterogeneity was noticed in some genes, such as Hexokinase Type I, Guanylate Cyclase Activator 1B, and Recoverin, which were reset in only a few numbers of iPSCs lines (16%) [113]. Using a standardized quantitative protocol called STEM-RET, the authors were able to score the capacity of the iPSCs, produced from the 5-cell type of the retina and from other sources such as fibroblasts, to differentiate into mature retinal structures [114]. Some cells, including immature bipolar neurons and rods, had the lowest reprogramming efficiency; however, they also had the highest STEM-RET score [113].

This epigenetic memory retained by the somatic donor cells has been used to produce iPSCs that could maintain their residual epigenome in order to enhance striatum fate differentiation. iPSCs derived from human whole ganglionic eminence (hWGE) were produced by Choompoo et al. and validated in their pluripotency. These hWGE iPSCs differentiate into neurons expressing a range of medium spiny neuron markers, similar to hESCs [115].

Similarly, Chlebanowska et al. noticed that, although the basic characteristics of the iPSCs did not significantly differ between different source cells, keratinocytes-derived iPSCs were more prone to induce the formation of higher numbers of neuroectodermal structures in comparison of fibroblast-derived iPSCs [116].

The differences found in cardiomyocytes are also interesting. Cardiomyocytes-derived iPSCs (iPSCs-CM) are already available and represent a reproducible and appropriate human cell source for cardiac disease modeling.

Pianezzi et al. compared cardiac-specific gene expression, Ca^2+^ properties, and electrophysiological characteristics of iPSC-CMs derived by reprogramming of adult human cardiac-derived mesenchymal progenitor cells (CPCs), bone marrow-derived mesenchymal stem cells MSCs (BMCs), and human dermal fibroblasts (HDFs) obtained from the same patients [117]. Since NK2 Homeobox 5 (Nkx2–5) is a key regulator of mesoderm differentiation to the heart, starting the synthesis of other transcription factors is typical of cardiomyocytes [118]. These authors suggest that a higher degree of functional maturity can be reached using cardiac somatic cell-derived iPSCs, with respect to non-cardiac cell sources. For instance, the beating capability is reached earlier from CM-iPSC with respect to HDF-iPSCs when differentiated [117].

Thus, epigenetic memory plays a decisive part in cell survival, as well differentiation, potentially affecting the outcome of the cellular model or the transplants by changing, respectively, the physiological properties and the adaptation of the grafted cells to the new environment.

## 7. Genetic Differences Drive Phenotypical Differences in iPSCs vs. ESCs

Up to now, we compared ESCs and iPSCs, primarily assuming that ESCs were the landmark with unvarying characteristics. This was despite the fact that a significant variation was observed for the differentiation efficiency of various human ES cell lines [119]. A high-throughput characterization of pluripotent cell lines, either ESCs or iPSC, allowed a final clarification of this diatribe between these pluripotent stem cells. Meissner et al. applied three genomic assays (DNA methylation mapping, gene expression profiling, and quantitative differentiation assay) to 20 ESC cell lines, and 12 iPSC cell lines established from many groups [120], providing a landmark of variation within human pluripotent cell lines. Due to this significant work, it is possible to score different pluripotent human stem cells for their capacity to work in a particular framework, such as studying neural function in vitro [120]. Underlying this finding was that none of the pluripotent cell lines were suitable for all the possible applications. Epigenetic and transcriptional variability between iPSCs and ESCs cannot be attributed exclusively to the somatic cell of origin, but other factors might account for it. Isogenic iPSCs and ESCs could not be discriminated by an analysis of their transcriptomes and differentiation potential, suggesting that genetic variability might be primarily responsible for the reported differences between iPSCs and ESCs [121,122]. When the global pattern of transcription from fibroblast, keratinocytes, and endothelial precursor-derived iPSCs was analyzed, the tissue of origin accounted for less than 4% of the transcriptional variation. In contrast, inter-individual genetic variation was responsible for 38% of the total, with <1% attributable to differences between iPSCs and ESCs [121]. Consistent with this observation, GpC methylation and differentially expressed gene analysis of fibroblast- or lymphoblast-derived iPSCs revealed little residual somatic memory, with genetic background being the major driver of the variation [123,124].

Donor-dependent variations were shown to affect differentiation potential and functional properties of iPSCs [122,124,125]. In their paper, Choi et al. found a high degree of similarity in methylation and gene expression. They noted a significant reduction in differentially expressed genes observed in the pluripotent state upon differentiation into fibroblast-like cells. This finding suggested that the differences in gene expression observed in the pluripotent state do not persist upon differentiation and might be compensated to become functionally irrelevant [122]. Differentiation into cells of the three germ layers revealed that no variations could be observed between genetically matched ESCs and iPSCs, thus confirming the relevance of donor effect on iPSCs’ differentiation potential [122]. Differences in gene transcription are maintained throughout differentiation and cluster with the donor rather than with the tissue of origin, causing a variable outcome upon differentiation [124]. More importantly, such differences are maintained after transplantation, impacting engraftment and differentiation potential [125] in vivo.

## 8. Conclusions

iPSC generation represented a significant breakthrough in biological sciences, offering a powerful tool for disease modeling, drug screening, personalized medicine, and tissue/organ regeneration, with no ethical issues. Although iPSCs were shown to possess, to some extent, various epigenetic and transcriptional differences compared to ESCs, these dissimilarities do not appear to have a functional impact on differentiation. Rather, donor-related variations represent a major source of variability that must be considered because it was shown to impact the phenotype of iPSCs and iPSCs-derived somatic cells.

The extent of the consequences of a donor effect must still be thoroughly studied and addressed before developing effective therapeutic strategies because these differences might be related to other aspects of the procedure linked also to the laboratory.

In a recent work [126], a robust meta-analysis approach that integrates RNA sequencing (RNA-seq) data and a genome-wide microarray was performed, indicating that there are at least three distinct naive-like pluripotent states that are found in the preimplantation embryo [127].

In conclusion, these differences are mostly due to interlaboratory data variability that comprises the cell source and the protocol. Thus, a great deal of work still needs to be conducted in order to increase the quality and homogeneity of these outcomes.

## Figures and Tables

**Figure 1 cells-10-01470-f001:**
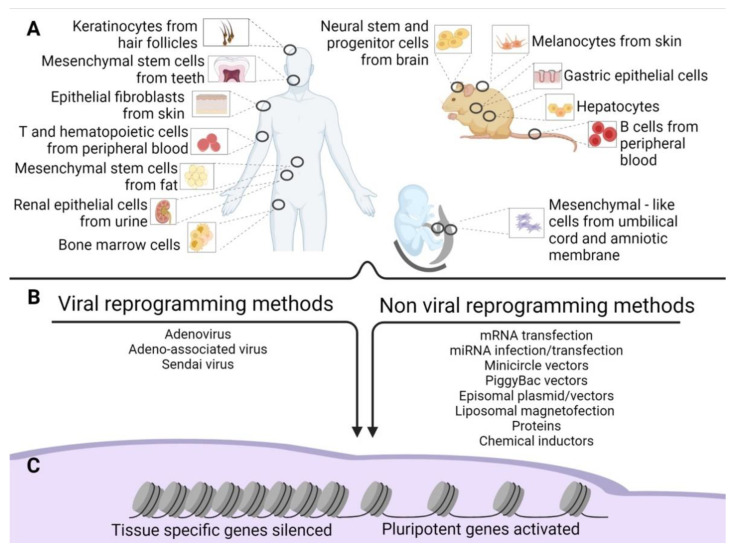
Old and new epigenetic memory in iPSCs. (**A**) Schematic representation of the possible tissue of origin of the source cells used for reprogramming in human and mouse adult tissues and extraembryonic human tissues. (**B**) Methods of reprogramming (viral and non-viral) of the source cells. (**C**) Gene silencing and activation after reprogramming.

**Figure 2 cells-10-01470-f002:**
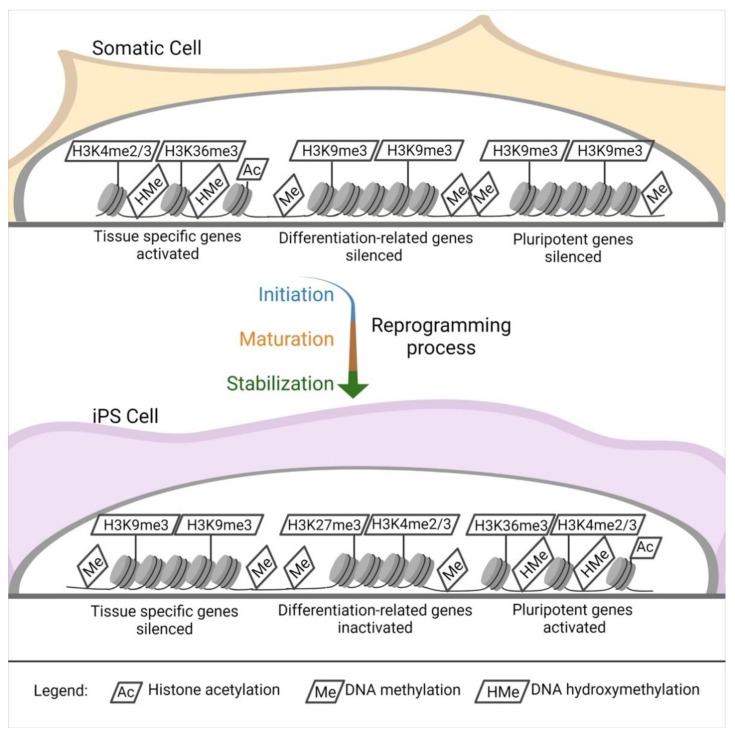
Epigenetic landscape changes. DNA and histones modification in the reprogramming process. Starting from a somatic cell throughout the reprogramming process (initiation, maturation, and stabilization), there is an intense modification of the histones and DNA in specific sites.

**Figure 3 cells-10-01470-f003:**
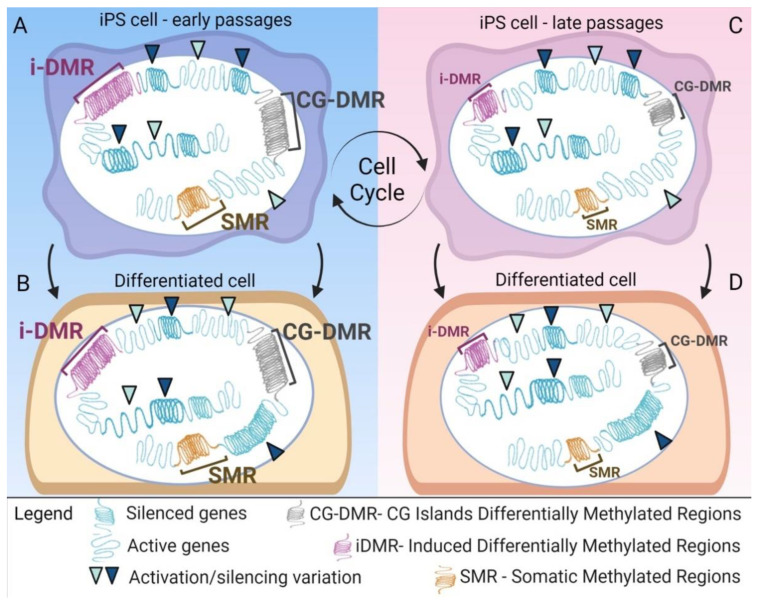
Old and new epigenetic memory in iPSCs. The figure summarizes how the modifications introduced by the reprogramming process fail to fully modify the epigenetic landscape of the original donor cell. Alongside the activation of genes that confer pluripotency characteristics, there are still states of activation, silencing, and methylation characteristic of the donor cell. (**A**) The persistence of methylated regions of the donor somatic cell, SMR (in brown), is found, in addition to the gene activation changes in the genes also related to pluripotency, cell cycling, cellular metabolism, differentiation, and tissue specificity, outlined by arrow points. Furthermore, at the end of the reprogramming process, there are also new aberrant methylation regions, induced (i)DMR (in violet), and methylations particularly concentrated in the CG islands—CG-DMR (in gray). (**B**) These new epigenetic arrangements remain even after the differentiation of the iPSCs into a new cell type. (**C**) With the expansion of iPSCs, there is a decrease in these differently methylated regions, which becomes clear at high passages. (**D**) This variation in the epigenetic landscape is also found in the differentiated cells.

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
