# Peer review of "iPSC Preparation and Epigenetic Memory: Does the Tissue Origin Matter?"

_cells, 2021, doi:10.3390/cells10061470_

Round 1

Reviewer 1 Report

In this review, the authors detail main methods for reprogramming source cells to generate iPSCs including potential tissues of origin and reprogramming efficiency. They then discuss the impact of the tissue of origin, the methods of reprogramming and the inter-individual genetic variation on the epigenetic alterations of the iPSCs and summarize main similarities and differences between ESCs and iPSCs.

The manuscript is timely, informative and gives an overview of a very interesting topic. Still, this reviewer has few concerns:

On the first sections discussing viral and non-viral reprogramming methods, I suggest to highlight or underline each method to make reading easier to follow up. I also suggest to expand the explanation of the details of each reprogramming approach.

Although the manuscript is overall well written, the narrative is at some points difficult to follow. Here I cite few examples but there are many throughout the text:

  • Line 65: These improvements merged approaches – which approaches are they referring to?
  • Line 252: following a stabilization phase is transgene independence – do they mean independent?
  • Line 313: variation of the epigenetic landscape is also found in the cells differentiated by them – cells differentiated by who?
  • Lines 355-356: the differences between ESCs and iPSCs could be depicted only comparing of biological replicated – I dont understand this sentence
  • Line 369-370: This methylation level typically expressed maternal Gtl2 in iPSCs aberrant from those ESCs – I dont understand this sentence
  • Line 389: SMP showed a high level of expression in the respective cell of origin than in the iPSCs – do they want to say higher?

I would therefore recommend reviewing the manuscript as the English style might not be adequate and there are few sentences difficult to understand.

I would invite the authors to make these amendments on their manuscript to make it more reader-friendly before it is ready to be published at Cells.

Author Response

Reviewer 1

In this review, the authors detail main methods for reprogramming source cells to generate iPSCs including potential tissues of origin and reprogramming efficiency. They then discuss the impact of the tissue of origin, the methods of reprogramming and the inter-individual genetic variation on the epigenetic alterations of the iPSCs and summarize main similarities and differences between ESCs and iPSCs.

The manuscript is timely, informative and gives an overview of a very interesting topic. Still, this reviewer has few concerns:

  1. On the first sections discussing viral and non-viral reprogramming methods, I suggest to highlight or underline each method to make reading easier to follow up. I also suggest to expand the explanation of the details of each reprogramming approach.

We underline each method to make reading easier to follow up. We expanded the explanation of the details of each reprogramming approach.

Viral reprogramming methods

Within the viral approaches, apart from the aforementioned genome-integrating vectors, we can include Adeno (AV), Adeno-associated (AAV), and Sendai viruses (SV).

Viral non-integrating reprogramming methods were developed to overcome concerns related to exogenous gene integration and DNA modification at the expense of generally lower reprogramming efficiency. For example, the adenoviral approach permits an efficiency of only 0.001 - 0.0001% in mouse fibroblasts [13] and 0.0002% in human fibroblasts [14]; multiple infections might be required [15]. In addition, the use of viral vectors might elicit an immune response in the host after cell transplantation, thus compromising the efficacy of the therapy [16].

Sendai virus

Sendai virus is a negative sense, mRNA virus belonging to Paramixoviridae family [17]. It is non-pathogenic to humans, and its use as a viral vector has several advantages: 1) being an mRNA virus, it does not enter the nucleus in its lifecycle, thus eliminating the risk of modifying the host genome and/or causing gene silencing by epigenetic changes [18]2) it shows a broad tropism, being able to infect several cell types in vitro [19-23] 3)due to its non-integrating nature, viral genome is diluted to every cell duplication, allowing its removal from the reprogrammed cells; and 4) it allows the production of a large number of proteins, thus allowing multiplicity of infection (MOI) reduction. Sendai viral vectors were successfully used to reprogram fibroblast [24] as well as blood [25,26] and renal epithelial cells in the urine [27]. This technique is quite efficient, ranging between 0.01 to 4% in the generation of human iPSCs at 25 days of induction [20,24,25,28,29]. Up to 10 passages or a high-temperature culture (39 °C) might be necessary to remove the viral genome completely [11]; however, recently it was developed auto-erasable, replication-deficient Senday virus by using microRNA-302 which impedes the viral replication by blocking the viral RNA-dependent RNA polymerase [30].

Adenovirus

Adenovirus is a non-integrating virus [31] that remains in the epichromosomal form in all cell types, except in egg cells [32]. Adenovirus offers a large cargo capacity, a transient expression, and rapid clearance from dividing cells, thus requiring multiple rounds of infection. The reprogramming efficiency is low, 0.001 - 0.0001% in mouse cells and 0.0002% in human cells most likely because of the low infection efficiency and the narrow expression window of reprogramming factors [16].

Adeno-associated virus

Adeno-associated virus is a non-pathogenic, non-autonomous single-stranded DNA virus, unable to replicate without the presence of a co-infecting helper virus. In its absence, AAV’s genome remains in episomal form within the infected cells, although integration into the host’s genome was reported in less than 10% of cases. AAVs are associated with weak immune reactions, primarily deriving from innate responses to the viral genome [33,34]. For all these reasons this vector has been used in more than 100 clinical trials [35]. However, the need for multiple rounds of transduction for cell reprogramming, the limited transgene capacity (5 kb), and the low efficiency (less than 0.01%) [16,35], still limit AAVs use as a vector for inducing pluripotency.

Non-viral reprogramming methods

Within these techniques, we can include mRNA transfection, miRNA infection/transfection, PiggyBac, Episomal plasmids/vectors, minicircle vectors, and proteins and chemical induction [11,31].

mRNA transfection

mRNA transfection was first used for cell reprogramming by Warren’s group. They overcame several obstacles to transcribe mRNAs to express reprogramming factors efficiently, reaching a 1.4 % reprogramming efficiency [36]. Moreover, the addiction of Lin28 to the Yamanaka reprogramming factor protocol, the valproic acid in the cell culture medium, and a change of O2concentration to 5% allowed an increase in efficiency to 4.4% [36]. The need of daily additions of mRNA to the cells limited to fibroblast make this method less appealing than other approaches [11].

miRNA infection/transfection

Regarding the miRNA infection/transfection, several miRNA such as miR-302b or miR-372 are strongly expressed in ESCs. Their addition to Yamanaka factors increased up to 15-fold reprogramming efficiency for OSKM alone [37]. Interestingly, some miRNAs could reprogram cells at high efficiency even in the absence of co-transfection with OSKM, bringing the efficiency of reprogramming for BJ-1 fibroblasts up to 10% [38].

Transposons: PiggyBac and Sleeping Beauty

These two transposons usually consist in a polycistronic transcript containing the OSKM factors joint by 2A peptides, allowing post-translational cleavage of the polyprotein into single reprogramming proteins as well as maintenance of their stoichiometric co-expression [39]. PiggyBac is a transposon, a mobile genetic element, that can be easily inserted and removed from chromosomal TTAA sites in the genome. Using a transposase, it can be integrated and subsequently excised from the genome [11]. When the OSKM factors are cloned into the PiggyBac vector and co-transfected into Mouse Embryonic Fibroblasts (MEFs),the reprogramming efficiency ranged from 0.02 to 0.05% of the total transfected cells [40,41]. This technique requires only a single transfection, the transposon can transport substantial cargo and presents low immunogenicity.

An intrinsic feature of the PiggyBac vector is its integration into the host genome. However, it could be cleanly excised from the iPSCs genome. Potential reintegration is conceivable due to the use of the same enzyme for insertion and excision. This reintegration risk forces a tight screening of iPSC clones to confirm the absence of integration and is time-consuming. As previously mentioned, reprogramming efficiency is quite low (0.01 - 0.1%) but can be improved using sodium butyrate. Regardless, the efficiency remains 50-fold lower than retroviral mediated reprogramming methods [42].

Sleeping Beauty transposon vector differs from PiggyBac for its capability to integrate randomly into the host genomes, thus showing no integration tendency with regards to specific genes and gene-regulatory elements [43].

Episomal plasmids

Episomal plasmids ensure technically simple procedure, a stable transgene expression due to their self-replication and low immunogenicity, allowing their removal by culturing cells in the absence of drug selection [44]. Due to their vulnerability to exonucleases, episomal vectors have an extremely low reprogramming efficiency, primarily due to the short expression time in the cells. It is possible to overcome this issue by repeating transfections daily; however, the reprogramming efficiency remains unsatisfactory (0.0003 - 0.0006%) [45]. The inclusion of NANOG, LIN28, and LT (SV40 large T gene) as reprogramming factors enhanced the efficiency 100 times, making it comparable to viral-based methods [46]. Instead of using a single plasmid for every Yamanaka factors which is laborious and less efficient as only a fews cells received all the plasmids, some groups used polycystronic plasmids to obtain a “3+1 delivery” of the reprogramming factors (with Oct4, Klf4, and Sox2, carried by one plasmid, and c-Myc by the other) [15], while other groups rely on one single plasmid to deliver all four genes [47], under the control of a constitutively active CAG promoter. However, these last methods do not ensure an adequate stoichiometric co-expression of the multiple reprogramming factors. The use of picornaviral 2A self-cleaving peptides to link reprogramming factors when is used a polycistronic construct [48] can partially ameliorate the balance of the expression of the four genes. However, there is still the chance that polycistronic plasmid could produce an unbalanced expression of each reprogramming factor [49]. This issue together with the large size of the plasmid hampers the efficiency of plasmid-based reprogramming [16].

Minicircle vectors

Minicircle vectors are supercoiled DNA episomal vectors similar to a standard plasmid but containing only the eukaryotic promoter and the cDNA(s) of the genes to be expressed. Their small size, resistance to cleavage, extremely low immunogenicity, and high transfection efficiency make them a good tool for cell reprogramming [16], despite a very low reprogramming efficiency (0.005%) and long reprogramming time (14–18 days). Thus, multiple rounds of transfections are required, causing a reduction of cell viability [50]. In order to improve the efficiency of reprogramming, various researchers used electropulsation, included additional reprogramming factors and/or microRNAs, used small molecules, and included hypoxic conditions [51-54]. Tight screening of the clones is necessary to exclude the integration of transgene sequences.

Liposomal magnetofection

Liposomal magnetofection is a non-viral technique that allows the delivery of nucleic acids in cultured cells by mixing nucleic acid and magnetic nanoparticles in cationic lipids. The lipids are concentrated on the surface of the cells using a magnetic field [55]. This technique has little chance of genomic integration, requires only a single transfection, and has low immunogenicity. There are rare cases of genomic integration. Consequently, screening iPSC clones is necessary to confirm the absence of integration [56]. Moreover, its reprogramming efficiency is between 0.032–0.040% after 8 days [16].

Proteins

Although the bioactive forms of reprogramming proteins can be synthesized by prokaryotic or eukaryotic systems, the main hurdle for reprogramming is their limited capability to cross the cell membrane. To overcome this obstacle the protein approach takes advantage of the HIV-TAT protein (a protein transduction domain) in delivering recombinant proteins. This technique allows the introduction of proteins into cells from the external environment without permeabilization agents [57]. The efficiency of this procedure is quite low at 0.006% of mouse fibroblasts [58] and 0.001% of human fibroblasts [9,59]. To improve the reprogramming efficiency some authors supplemented the culture media with Valproic Acid (VPA), with 0.006% of cells induced to pluripotency after 30-35 days [58].

Protein transduction domains later became a method to deliver not only proteins but also other macromolecules. Those included peptide nucleic acids (PNA), antisense, short-interfering ribonucleic acids (siRNA), liposomes, iron nanoparticles, and plasmids [57].

Chemical induction

Despite the methods mentioned above for reprogramming, the highest degree of safety at the cost of low reprogramming efficiency is represented by iPSC generation through the use of small molecules to obtain chemically induced pluripotent stem cells (CiPSCs).

Hou et al. developed a combination of six small molecules (obtained after an intense work of screening of more than 10,000 compounds). They included several cAMP agonists (Forskolin, Rolipram, and Prostaglandin E2) and epigenetic modulators (sodium butyrate, 3-deazaneplanocin A (DZNep), 5-Azacytidine, and RG108) to generate chemically induced iPSCs (CiPSCs). Interestingly, they found that small molecules (sm) iPSCs could be generated using only one gene of the OSKM, namely Oct4, with the addiction of CHIR99021, tranylcypromine (VC6T), VPA, and 616452 [8,60]. Compared to ESCs, CiPSCs have similar doubling time, gene expression profiles, differentiation ability, generate teratomas and chimeric mice [8,61]. Moreover, it is intriguing that different chemical cocktails are needed to induce other source cells [60]. Up to date, the current chemical reprogramming efficiency is only 0.2% with an induction time of more than 36 days that were recently reduced up to 16 days [62,63].

Although the manuscript is overall well written, the narrative is at some points difficult to follow. Here I cite few examples but there are many throughout the text:

  • Line 65: These improvements merged approaches – which approaches are they referring to?

The sentence was changed as:

These changes in reprogramming methods merged approaches still based on integration into the host genome of the exogenous reprogramming factors and included lentiviruses and transposons as vectors.

  • Line 252: following a stabilization phase is transgene independence – do they mean independent?

Reviewer 1 is correct independence was changed.

  • Line 313: variation of the epigenetic landscape is also found in the cells differentiated by them – cells differentiated by who?

The sentence was changed: D) This variation of the epigenetic landscape is also found in the differentiated cells.

  • Lines 355-356: the differences between ESCs and iPSCs could be depicted only comparing of biological replicated – I don’t understand this sentence

The sentence was rephrased:

In contrast, the differences between ESCs and iPSCs could be depicted only when more biological replicated were added, most likely because this variability was introduced by the inclusion of different donors

  • Line 369-370: This methylation level typically expressed maternal Gtl2 in iPSCs aberrant from those ESCs – I dont understand this sentence

Indeed, this sentence was mistaken, and it was eliminated

  • Line 389: SMP showed a high level of expression in the respective cell of origin than in the iPSCs – do they want to say higher?

Reviewer 1 is correct, we modified the word

I would therefore recommend reviewing the manuscript as the English style might not be adequate and there are few sentences difficult to understand.

The text was revised for English style by a collogue from the US.

Reviewer 2 Report

Scesa et al. review here in great details the methods to derive iPSC and their efficiencies vs drawbacks. Then they briefly conclude that the method used to derive iPSCs does not really change the quality of iPSCs obtained. They then ask whether the tissue of origin matters, but again conclude that the effects are minor. Finally, they leave the reader with the idea that variations in the genetic background of donor cells is what accounts for most of the difference in iPSC quality. Overall an interesting piece because it puts together several publications to extract these 3 main interesting conclusions. However, it still needs a few fixes (outlined below) before being considered for publication.

Major points:

  • Authors spend lots of efforts (5+ pages) enumerating the possible methods to make iPSCs, their efficiencies and drawbacks, yet they only very briefly summarize that no significant difference exist in isogenic iPSCs generated via different methods (line 189). That main conclusion should be made clearer and placed at the end of the section, so that it stands out. Importantly, are there any other papers than ref. 41 (which dates from 2014) that would support this idea? The argument feels a bit thin.
  • The main subject referred in the title is only discussed for about 1 page in the paper (lines 380-439). I wonder whether the title could be made more representative of the content, and/or whether that section could be expanded (see next point).
  • All of the papers cited in the last two sections – the most interesting ones – were published in or before 2016. Is there really nothing that has been done in the field in the past 5 years that would push the theories put forth in the review one side or another? These two sections need to be expanded and cite newer papers.
  • Authors conclude the paper saying that “donor variations represent a major source of variability impacting the phenotype of iPSC”. Could they state more clearly whether it means that cells from certain individuals may yield better quality iPSCs than cells from others?

Minor points:

  • A typo is present in multiple figures: “pluripontent”.
  • Line 13: “concerns concerning”.
  • Lines 16-18: sentence is confusing and is contradicted later in the paper. I would suggest to replace by “… cell source origin may be pivotal to iPSC generation since residual epigenetic memory could influence the iPSC phenotype and transplantation outcome” or equivalent.
  • The abstract is about humans, but the 1st paragraph switches to work done in mice without a warning. Please specify.
  • Line 39: Please summarize the scheme proposed by Thomson so the reader does not have to read that paper to understand your point.
  • Line 41: sentence not accurate; “Yamanaka’s share of the Nobel prize…”
  • Line 178: define smiPSCs.
  • Line 204: “cells […] might have the power to be converted” - or rephrase!
  • Line 216: replace “Despite” by “Independently of” or equivalent.
  • Line 261: sentence is unclear; what is the meaning of putting “de” in parentheses? Do you mean methylated or demethylated, or a mix of both?
  • Line 266: “began” needs to be removed.
  • Line 270: Add “NSC-derived” before “iPSCs”.

Author Response

Reviewer 2

Scesa et al. review here in great details the methods to derive iPSC and their efficiencies vsdrawbacks. Then they briefly conclude that the method used to derive iPSCs does not really change the quality of iPSCs obtained. They then ask whether the tissue of origin matters, but again conclude that the effects are minor. Finally, they leave the reader with the idea that variations in the genetic background of donor cells is what accounts for most of the difference in iPSC quality. Overall an interesting piece because it puts together several publications to extract these 3 main interesting conclusions. However, it still needs a few fixes (outlined below) before being considered for publication.

Major points:

  • Authors spend lots of efforts (5+ pages) enumerating the possible methods to make iPSCs, their efficiencies and drawbacks, yet they only very briefly summarize that no significant difference exist in isogenic iPSCs generated via different methods (line 189). That main conclusion should be made clearer and placed at the end of the section, so that it stands out. Importantly, are there any other papers than ref. 41 (which dates from 2014) that would support this idea? The argument feels a bit thin.

The paragraph was changed accordingly to the Reviewer 2 suggestions.

Comparison between different methods of reprogramming

The presence of the high level of copy number variation (CNV) in hiPSCs compared to hESCs or human somatic cell samples can be explained with two, not self-excluding, hypotheses; a) they are gained de novo during the reprogramming procedure or in vitro iPSCs culture or, b) they are present in the starting somatic cell population that also could be a mosaic [64-66]. Since the first work of Yamanaka [3], many efforts have been made to understand how reprogramming could impact the quality of the iPSCs.

In a study conducted by Ma et al., the comparison between different methods of reprogramming (i.e. Sendai virus (IPSCs-S) and retroviral (iPSCs-R) methods) indicated that some lines, such as iPS-S4, iPS-S5, and iPS-R2, did not display significant genomic macroscopic alterations. Copy number variation (CNV) analysis did not entirely exclude the presence of small insertion-deletion (indels), point mutations, or translocations [67].

In other papers, the genetic stability of independent iPSCs lines with common donors was tested by CNV SNP microarrays [65]. However, lines produced using integrating vector showed a trended but not significantly higher frequency of clinically significant CNV (58%) compared with non-integrating vectors (41%). Since this study compared iPSCs lines obtained from the same donor the authors could evaluate if the CNV differences were due to the tissue of origin or the method or reprogramming [65]. Similarly, Schlaeger et al, [29] compared episomal vector reprogramming, Sendai virus, RNA, and lentivirus reprogramming finding no differences in CNV. Many different groups found that if differences do exist between the reprogramming methods these are most likely present when the reprogramming is made using integrating viral vectors and are very subtle although could be more deleterious [68-71].

Take together, these data suggest that different induction methods do not contribute significantly to the genic alterations found in iPSC lines obtained from isogenic cells; most likely the genic impairment found in iPSCs could be ascribed to the somatic donor cells or the cultivation time.

Source cells for reprogramming

The cells used for reprogramming depend on the organism, the availability of the tissue, and the kind of differentiated cellswe would like to produce from the iPSCs. As addressed in the following sections, many tissues cannot be used because they are unavailable (i.e., brain tissue) unless obtained as discarded tissue.

However, to date, iPSCs have been obtained using a plethora of tissues (Fig. 1B). In this regard, some of these methods require invasive procedures such as biopsies, as in primary skin fibroblasts. More accessible sources are available, namely peripheral blood from which retrieve T cells [25], B cells [72], hematopoietic stem cells [73], and bone marrow cells. Recently, iPSCs have been produced, choosing less invasive cells to obtain, such as keratinocytes isolated from hair follicles [74]. Very often, cell sources have been obtained from biological waste material [75]. Examples include renal epithelial cells in the urine [27,76], mesenchymal stem cells from teeth and fat tissue [77], liver and stomach cells [78], melanocytes [79], neural stem and progenitor cells [80], and embryonic and extraembryonic tissue [81]. These outcomes indicate that cells of all tissues might be converted into iPSCs. The final point about the origin of source cells concerns their age. Senescent cells or cells obtained from the elderly are induced to iPSCs with more difficulties. However, Lapasset et al. found that their induction efficiency could be increased using a six-factor reprogramming cocktail (SOX2, OCT4, KLF4, NANOG, LIN28, and c-MYC) instead of the usual OSKM, also eliminating the marks of cellular aging [82,83].

Depending on the cells of origin and the methods of reprogramming, other genes cocktail than OSKM, such as p53 shRNA, Lin28, L-Myc, SV40LT, Nanog, Glis1, and others have been used in different reprogramming mixes, sometimes improving the efficiency of reprogramming, in particular subsets of tissue sources [27,84].

A very important aspect that we mentioned early is the presence of mosaicisms in the source cells that could negatively affect the reprogrammed cells. The production of iPSCs from a patient affected by Down syndrome showed that the patient was a mosaic since one-third of the reprogrammed cell were euploid whereas the remaining 66% were trisomic [85]. Abyzov et al. demonstrated that 50% of the CNVs identified in the hiPSC lines were detectable, even if at a very low frequency, in the source fibroblast population [66] indicating the presence of somatic mosaicism in these cells.

  • The main subject referred in the title is only discussed for about 1 page in the paper (lines 380-439). I wonder whether the title could be made more representative of the content, and/or whether that section could be expanded (see next point).

The section was expanded (almost doubled) according to Reviewer 2 indications more and newer references were added.

Hargus et al. differentiated human iPSCs derived from fetal NSCs and dermal fibroblasts into neural precursor cells (NPCs) [112]. Whole-genome and brain-specific gene expression analysis revealed clear segregation of the different cell groups according to their somatic origin and retention of specific transcripts due to epigenetic memory. Interestingly, the authors assessed the functional effect of epigenetic memory of NPCs upon transplantation. However, the percentage of iPSCs-derived neurons was similar between the two groups after 5 weeks of in vitro differentiation. NPCs from fNSCs-iPSCs yielded a higher number and more differentiated neuronal cells than Fib-iPSCs-derived NPCs [112] when transplanted in cortex and striatum indicating that the neural niche has more favorable effects for neuroectodemal than mesodermal iPSC-derived neurons.

On the other side, many efforts have been made to search for the appropriate tissue of origin to produce iPSCs to be differentiated into mature cells of a particular lineage, either for physiological study or for transplantation. Neurons and cardiomyocytes have been frequently subject of these studies.

In recent work, Wang et al. studied the epigenetic memory of reprogramming efficiency of five different retinal cell types [113]. The authors determine whether r-iPSCs retain an epigenetic memory by selecting 33 lines (comprising iPSCs from 5 retinal cell types and fibroblast-derived iPSCs). A thorough study with different techniques including DNA methylation analysis, RNA-seq, and ChIP-seq of 8 histone marks was performed. Some progenitor genes including Nestin, Paired Box6, Sine Oculis Homeobox Homolog 3 and 6, Visual System Homeobox 2, and Meis Homeobox 1 showed DNA methylation profiles that was reset to the ESC state in almost all the iPSC lines. Some heterogeneity was noticed in some genes, such as Hexokinase Type I, Guanylate Cyclase Activator 1B, and Recoverin, which were reset in only a few numbers of iPSCs lines (16%) [113]. Using a standardized quantitative protocol called STEM-RET the authors were able to score the capacity of the iPSCs, produced from the 5-cell type of the retina and from other sources such as fibroblasts, to differentiate into mature retinal structures [114]. Some cells including immature bipolar neurons and rods had the lowest reprogramming efficiency, however, they have the highest STEM-RET score [113].

This epigenetic memory retained by the somatic donor cells has been used to produce iPSCs that could maintain their residual epigenome in order to enhance striatum fate differentiation. iPSCs derived from human whole ganglionic eminence (hWGE) were produced by Choompoo et al. and validated in their pluripotency. These hWGE iPSCs differentiate into neurons expressing a range of medium spiny neuron markers similarly to hESCs [115].

On the same line, Chlebanowska et al. noticed that, although the basic characteristics of the iPSCs did not significantly differ between different source cells, keratinocytes-derived iPSCs were more prone to induce the formation of higher numbers of neuroectodermal structures in comparison of fibroblasts derived iPSCs [116].

Very interesting are also the differences found in cardiomyocytes. Cardiomyocytes derived iPSCs (iPSCs- CM) are already available and represent a reproducible and appropriate human cell source for cardiac disease modeling.

Pianezzi et al. compared cardiac-specific gene expression, Ca2+ properties, and electrophysiological characteristics of iPSC-CMs derived by reprogramming of adult human cardiac-derived mesenchymal progenitor cells (CPCs), bone marrow-derived mesenchymal stem cells MSCs (BMCs), and human dermal fibroblasts (HDFs) obtained from the same patients[117]. Since NK2 Homeobox 5 (Nkx2–5) is a key regulator of mesoderm differentiation to the heart starting the synthesis of other transcription factors typical of the cardiomyocytes [118]. These authors suggest that a higher degree of functional maturity can be reached using cardiac somatic cell-derived iPSCs, with respect to non-cardiac cells source. For instance the beating capability is reached earlier from CM-iPSC with respect to HDF-iPSCs when differentiated [117].

Thus, epigenetic memory plays a decisive part in cell survival as well differentiation, potentially affecting the outcome of the cellular model or the transplants by changing respectively the physiological properties and the adaptation of the grafted cells to the new environment.

  • All of the papers cited in the last two sections – the most interesting ones – were published in or before 2016. Is there really nothing that has been done in the field in the past 5 years that would push the theories put forth in the review one side or another? These two sections need to be expanded and cite newer papers.

Reviewer 2 made an important point. Indeed, as mentioned in the previous point the total number of references was increased from 84 to 127, and the references from 2017 to 2021 were increased from 14 to 26 in particular in the aforementioned sections

  • Authors conclude the paper saying that “donor variations represent a major source of variability impacting the phenotype of iPSC”. Could they state more clearly whether it means that cells from certain individuals may yield better quality iPSCs than cells from others?

The conclusion was revised and clarified.

iPSC generation represented a significant breakthrough in the biological sciences, offering a powerful tool for disease modeling, drug screening, personalized medicine, and tissue/organ regeneration with no ethical issues. Although iPSCs were shown to possess, to some extent, various epigenetic and transcriptional differences compared to ESCs, these dissimilarities do not appear to have a functional impact on differentiation. Rather, donor-related variations represent a major source of variability that must be considered because it was shown to impact the phenotype of iPSCs and iPSCs-derived somatic cells.

The extent of the consequences of a donor effect must still be thoroughly studied and addressed before developing effective therapeutic strategies because these differences might be related to other aspects of the procedure linked also to the laboratory.

In a recent work [126], a very robust meta-analysis approach that integrates RNA sequencing (RNA-seq) data and genome-wide microarray was performed and indicated that there are at least three distinct naive-like pluripotent states that are found in the preimplantation embryo [127].

In conclusion, these differences are mostly due to interlaboratory data variability that comprises the cell source and the protocol so still a huge work has to be done to increase the quality and homogeneity of the outcome.

Minor points:

  • A typo is present in multiple figures: “pluripontent”.

The typos were corrected in all the figures.

  • Line 13: “concerns concerning”.

Concerning was changed with regarding

  • Lines 16-18: sentence is confusing and is contradicted later in the paper. I would suggest to replace by “… cell source origin may be pivotal to iPSC generation since residual epigenetic memory could influence the iPSC phenotype and transplantation outcome” or equivalent.

The sentence was changed according to Reviewer 2 suggestion.

On the other hand, cell source origin may be pivotal to iPSC generation since residual epigenetic memory could influence the iPSC phenotype and transplantation outcome.

  • The abstract is about humans, but the 1stparagraph switches to work done in mice without a warning. Please specify.

The first sentence of the introduction was modified, and the appropriate reference was introduced

  • Line 39: Please summarize the scheme proposed by Thomson so the reader does not have to read that paper to understand your point.

Thomson paper was summarized

This opportunity marks a decisive step compared to the generation of human ESCs arranged by Thomson et al. in 1998 [2]where they described the derivation of pluripotent stem cells from the human blastocyst and depicted their characteristics such as retention of normal karyotypes, high expression levels of telomerase activity, high proliferation, and differentiation capability.

  • Line 41: sentence not accurate; “Yamanaka’s share of the Nobel prize…”

Reviewer 2 is correct: the sentence was changed:

2012 Nobel prize sanctioned that specialization of cells is reversible and that adult somatic cells could be reprogrammed to an immature, pluripotent state.

  • Line 178: define smiPSCs.

The term was defined: small molecules (sm) iPSCs

  • Line 204: “cells […] might have the power tobe converted” - or rephrase!

The sentence was changed according to Reviewer 2 suggestion:

These outcomes indicate that cells of all tissues might be converted into iPSCs

  • Line 216: replace “Despite” by “Independently of” or equivalent.

The term was changed following Reviewer 2 suggestion

  • Line 261: sentence is unclear; what is the meaning of putting “de” in parentheses? Do you mean methylated or demethylated, or a mix of both?

Indeed the (de) between parentheses could be misleading and we changed: Finally, DNA methylation and demethylation increases during the stabilization phase [87,88]

  • Line 266: “began” needs to be removed.

The term was removed

  • Line 270: Add “NSC-derived” before “iPSCs”.

NSC-derived was introduced

Reviewer 3 Report

This review article introduced the history of iPS generation, how they have been improved, and also referred to epigenetic memory. The authors would like to declare not somatic origin but intragenetic difference would impact the iPSC differences. This review is very interesting and can learn the importance to study the epigenic differences for efficient regenerative medicine.

Concerns

Lines 137-140, Although the transposon integrates into the genome, it is easily excisable. An intrinsic feature of the PiggyBac vector is its integration into the host genome. However, it could be cleanly excised from the iPSCs genome.  →The authors had better not repeat the same content.

Lines 264-266, Initially, iPSCs were considered fully equivalent to ESCs, given their similarity in mor-phology, proliferation, markers expression, differentiation potential, and teratomas for-mation [1]. →The authors of the referenced paper did not indicate iPSCs were fully equivalent to ESCs. The electrophoresis gel images and methylation analysis of iPSCs in that paper showed similarity to ESCs and difference to the parental MEFs.

Figure 3 is difficult to understand. The authors should make a more clearer figure.

Line 322, What is iDMR?

Lines 355-356, In contrast, the differences between ESCs and iPSCs could be depicted only comparing of biological replicated, most likely because this variability was introduced by the inclusion of different donors [69]. →This sentence is difficult to understand. I think it is better to describe in simple sentences.

Lines 368-369, These include an intergenic DMR (IG-DMR) and intragenic differential methylation at the level of Glt2 promoter [71,72]. →I cannot understand “at the level of”

Line 370, aberrant from those ESCs. →aberrant from those of ESCs. (?)

Lines 371-378, Very interestingly, ESCs (containing the OKSM transgene) and iPSCs derived from MEFs from the same genetic background were compared for their capacity to support the development of all-iPSC mice using 4n embryo complementation. ESC cell lines produced viable mice at the anticipated frequencies (13 - 20%), proving that the OKSM  transgene does not invalidate embryo development. On the other hand, all the tested iPSC voided to do so [68]. →This experiment is difficult to understand. Especially all-iPSC.

Lines 433-436, However, the percentage of iPSCs-derived neurons was similar between the two groups. NPCs from fNSCs-iPSCs yielded a higher number and more 435 differentiated neuronal cells than Fib-iPSCs-derived NPCs [77]. →Confusing. Which sentence is correct?

Author Response

Reviewer 3

This review article introduced the history of iPS generation, how they have been improved, and also referred to epigenetic memory. The authors would like to declare not somatic origin but intragenetic difference would impact the iPSC differences. This review is very interesting and can learn the importance to study the epigenic differences for efficient regenerative medicine.

Concerns

  • Lines 137-140, Although the transposon integrates into the genome, it is easily excisable. An intrinsic feature of the PiggyBac vector is its integration into the host genome. However, it could be cleanly excised from the iPSCs genome.  →The authors had better not repeat the same content.

The sentence “Although the transposon integrates into the genome, it is easily excisable” was removed

  • Lines 264-266, Initially, iPSCs were considered fully equivalent to ESCs, given their similarity in mor-phology, proliferation, markers expression, differentiation potential, and teratomas for-mation [1]. →The authors of the referenced paper did not indicate iPSCs were fully equivalent to ESCs. The electrophoresis gel images and methylation analysis of iPSCs in that paper showed similarity to ESCs and difference to the parental MEFs.

The sentence was changed as:

 Initially, iPSCs were considered very similar to ESCs, given their similarity in morphology, proliferation, markers expression, differentiation potential, and teratomas formation [3].

  • Figure 3 is difficult to understand. The authors should make a more clearer figure.

Figure 3 was changed by introducing a symbol arrowhead that improved the readability of the figure, the different sizes of the letters also indicated the changes in methylation levels.

  • Line 322, What is iDMR?

The definition of iDRM was introduced: induced DMR

  • Lines 355-356, In contrast, the differences between ESCs and iPSCs could be depicted only comparing of biological replicated, most likely because this variability was introduced by the inclusion of different donors [69]. →This sentence is difficult to understand. I think it is better to describe in simple sentences.

The sentence was changed:
Similar outcomes were described in a study showing an analogous result at the proteome level. In contrast, the differences between ESCs and iPSCs could be depicted only when more biological replicated were added, most likely because this variability was introduced by the inclusion of different donors [100]

  • Lines 368-369, These include an intergenic DMR (IG-DMR) and intragenic differential methylation at the level of Glt2 promoter [71,72]. →I cannot understand “at the level of”

The sentence was modified:

These include an intergenic DMR (IG-DMR) and intragenic differential methylation in the Glt2 promoter [102,103]

  • Line 370, aberrant from those ESCs. →aberrant from those of ESCs. (?)

We apologized for this mistake, the sentence was removed

  • Lines 371-378, Very interestingly, ESCs (containing the OKSM transgene) and iPSCs derived from MEFs from the same genetic background were compared for their capacity to support the development of all-iPSC mice using 4n embryo complementation. ESC cell lines produced viable mice at the anticipated frequencies (13 - 20%), proving that the OKSM  transgene does not invalidate embryo development. On the other hand, all the tested iPSC voided to do so [68]. →This experiment is difficult to understand. Especially all-iPSC.

The sentence was rephrased, and more explaining references were added:

The fibroblasts differentiated from iPSCs that acquired a silenced paternal methylation state did not activate after the induction of the differentiation induced by valproic acid [99]. Very interestingly, ESCs (containing the OKSM transgene) and iPSCs derived from MEFs from the same genetic background were compared for their capacity to support the development of entirely iPSC-derived animals (all-iPSC mice) using 4n embryo complementation, a technique in which 2n pluripotent cells are injected into tetraploid (4n) blastocyst. The resulting 2n-4n chimeric embryos will develop normally, with the extraembryonic tissues composed of 4n cells, while the fetus will originate from the injected 2n ESCs [104-106]. ESC cell lines produced viable mice at the anticipated frequencies (13 - 20%), proving that the OKSM transgene does not invalidate embryo development. On the other hand, all the tested iPSC voided to do so [99].

  • Lines 433-436, However, the percentage of iPSCs-derived neurons was similar between the two groups. NPCs from fNSCs-iPSCs yielded a higher number and more 435 differentiated neuronal cells than Fib-iPSCs-derived NPCs [77]. → Which sentence is correct?

The sentence was changed:

However, the percentage of iPSCs-derived neurons was similar between the two groups after 5 weeks of in vitro differentiation. NPCs from fNSCs-iPSCs yielded a higher number and more differentiated neuronal cells than Fib-iPSCs-derived NPCs [112]when transplanted in cortex and striatum indicating that the neural niche has more favorable effects for neuroectodemal than mesodermal iPSC-derived neurons.

On the other side, many efforts have been made to search for the appropriate tissue of origin to produce iPSCs to be differentiated into mature cells of a particular lineage, either for physiological study or for transplantation. Neurons and cardiomyocytes have been frequently the subject of these studies.

Round 2

Reviewer 2 Report

The authors have satisfactorily addressed all comments.

Reviewer 3 Report

I think the ms improved significantly.